# Reliability-Guided Gradient Correction for Visible-Infrared Object Detection

## Abstract

Visible-infrared object detection has attracted increasing attention for its ability to fuse complementary information from visible and infrared sensors. While such fusion improves detection accuracy and robustness, it remains vulnerable to semantic conflicts due to inconsistent object representations across modalities. Existing works typically address these conflicts by aligning cross-modal features or adjusting modality weights using heuristic cues. However, they often overlook modality reliability, which reflects how well each modality captures object-relevant information, resulting in performance drops when unreliable features are used. To address this, we introduce RaGrad, a model-agnostic method for **r**eli**a**bility-guided **grad**ient correction to mitigate cross-modal semantic conflicts. Specifically, we first propose the **r**eliability **e**stimation via **p**arameter **a**ttribution (REPA) module, which estimates the reliability of modality-specific parameters by evaluating their effectiveness via counterfactual reasoning and sensitivity via gradient variation. Second, we propose the **r**eliability-**g**uided **c**onflict **r**esolution (RGCR) module, which resolves cross-modal conflicts by correcting the gradients of less reliable modalities under the guidance of more reliable ones, thereby promoting the learning of more reliable features and enhancing cross-modal consistency. Extensive experiments on three challenging datasets demonstrate the efficacy and generalizability of RaGrad, consistently improving performance across various baselines.

## 1 Introduction

Visible-infrared object detection improves both accuracy and robustness by leveraging the complementary information between modalities (Zhang et al., 2023d; Zeng et al., 2024). For example, visible (RGB) images provide rich semantic details under favorable lighting conditions but struggle in low-light or complex environments (Rothmeier et al., 2023). In contrast, infrared (IR) sensors capture stable thermal signals that remain consistent regardless of lighting conditions, thereby enhancing detection performance across a wide range of challenging scenarios (Sun et al., 2024).

Despite the benefits of RGB-IR fusion, intrinsic modality differences often result in inconsistent representations of the same object in RGB and IR images, thereby causing semantic conflicts (He et al., 2023). These conflicts degrade both localization and classification accuracy, ultimately limiting the performance and robustness of multimodal detection systems (Fu et al., 2024; Bao et al., 2025). To address this, recent studies have explored two main strategies. One approach focuses on aligning modality-specific representations before fusion by leveraging relevant features from the other modality to promote semantic consistency (Zhu et al., 2023; Tian et al., 2024; Yuan & Wei, 2024). Another approach dynamically adjusts modality importance based on heuristic cues such as illumination intensity, emphasizing the more informative modality under specific conditions to mitigate inconsistencies (Zhang et al., 2023c; Hu et al., 2025; Shang et al., 2025).

However, most existing methods overlook a critical issue: the features selected by the model may not always be reliable (Zhang et al., 2025), often manifesting as insufficient attention to object-relevant regions or emphasis on irrelevant areas. Such unreliable features may mislead the fusion process, impairing fusion quality and ultimately degrading overall detection accuracy and robustness. Fig. 1 shows how unreliable modality features compromise detection. In CALNet, the RGB features fail to attend to the "other" target (red dashed circle), whereas the IR features exhibit modest attention. However, the unreliable RGB features interfere with the fusion process and in turn

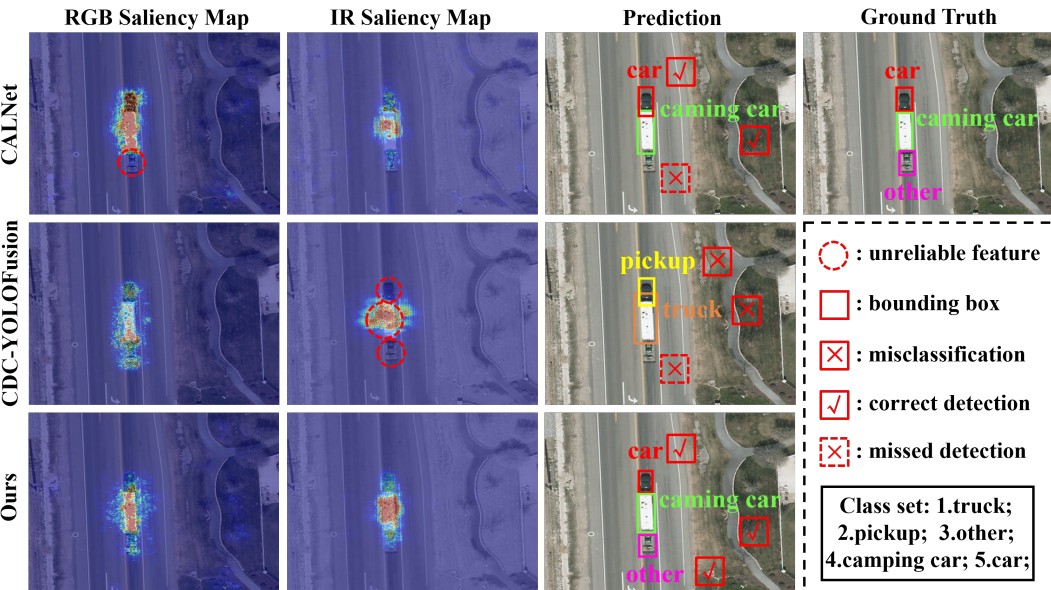

Figure 1: Comparison of modality-specific saliency maps and predictions across CALNet He et al. (2023), CDC-YOLOFusion Wang et al. (2024c), and our method. IR and RGB saliency maps are generated via gradient saliency Simonyan et al. (2014), where brighter regions indicate stronger focus on modality-specific features.

lead to the missed detection of the "other" target. In CDC-YOLOFusion, the RGB features reliably capture both the "camping car" and "other" targets. In contrast, the IR features are unreliable, as they poorly represent the "camping car" and fail to capture the "other" target, which compromises the fused representation, resulting in misclassifying the "camping car" as "truck" and missing the "other" target entirely. In comparison, our method maintains reliable and consistent features across both modalities, ensuring accurate classification and localization. These cases demonstrate that the unreliable features critically undermine detection performance.

Therefore, learning reliable modality features is essential for robust and accurate detections. Since these features are learned through their corresponding parameters, we assess modality reliability, which reflects how well each modality captures object-relevant information, by evaluating parameter behavior during training. This reliability information is then used to guide parameter updates, promoting the learning of more reliable modality-specific features. promoting the learning of more reliable modality-specific features.

Based on the above insight, we propose RaGrad, a model-agnostic method that performs **r**eli**a**bility-guided **grad**ient correction to enhance feature reliability across modalities, thereby alleviating cross-modal semantic conflicts in visible-infrared object detection. RaGrad consists of two key components. First, we propose the **r**eliability **e**stimation via **p**arameter **a**ttribution (REPA) module, which quantifies the reliability of modality-specific parameters by jointly evaluating their effectiveness via counterfactual reasoning and sensitivity via gradient variation, yielding a comprehensive reliability estimate. Second, we introduce the **r**eliability-**g**uided **c**onflict **r**esolution (RGCR) module, which resolves cross-modal conflicts by performing reliability-guided gradient correction. RGCR refines the parameter updates of less reliable modalities using the gradients of more reliable ones as guidance, thereby improving the consistency between modalities and enhancing overall detection performance. We validate RaGrad by integrating it into multiple detection frameworks on three challenging benchmark datasets, including VEDAI (Razakarivony & Jurie, 2016), LLVIP (Jia et al., 2021), and DroneVehicle (Sun et al., 2022). The results consistently show improved performance across all baseline methods. Our main contributions are summarized as follows:

- We propose RaGrad, a model-agnostic method for **r**eli**a**bility-guided **grad**ient correction to alleviate cross-modal semantic conflicts, which can be applied to most existing visible-infrared object detection frameworks.

- We propose the reliability estimation via parameter attribution module, which evaluates modality reliability by assessing the effectiveness and sensitivity of model parameters, providing a more accurate basis for cross-modal optimization.

- We propose the reliability-guided conflict resolution module, which resolves cross-modal conflicts by leveraging the gradients of reliable modalities to guide parameter updates, improving consistency and detection performance.

- We conduct comprehensive experiments across multiple detection frameworks on three challenging datasets, validating the efficacy and generalizability of our method.

## 2 RELATED WORK

**Visible-Infrared Object Detection.** Visible-infrared object detection benefits from the complementary information of RGB and IR modalities, but suffers from semantic conflicts due to inconsistent object representations across modalities. To address this, prior works have mainly explored two strategies. The first focuses on refining modality-specific features before fusion by leveraging cross-modal contextual similarities to enhance semantic consistency (He et al., 2023; Yuan & Wei, 2024; Chen et al., 2024). The second dynamically weights modalities based on heuristic cues like illumination to guide feature fusion (Zhang et al., 2019; Lai et al., 2023; Hu et al., 2025; Shang et al., 2025). However, they overlook that the selected modality features may be unreliable, which can degrade performance. In contrast, we estimate the modality reliability and utilize this information to refine parameter optimization, facilitating the learning of more reliable features.

**Gradient Correction.** During training, gradients from different modalities may conflict, hindering optimization and degrading performance. This issue has been well studied in multi-task learning (Yu et al., 2020; Liu et al., 2023; Chen & Er, 2025) and has also emerged in multimodal learning (Wu et al., 2022; Hua et al., 2024). Existing methods address this by removing conflicting components or adjusting gradients based on modality reliability or convergence speed (Wang et al., 2024a; Lin et al., 2024). However, these methods generally rely on coarse conflict detection criteria, which may lead to unnecessary adjustments, potentially resulting in suboptimal corrections (Liu et al., 2021a). In contrast, we jointly consider gradient direction and modality reliability to better identify significant conflicts and perform reliability-guided correction for more efficient and robust optimization.

## 3 METHOD

**Overview.** To alleviate semantic conflicts in visible-infrared object detection, we propose RaGrad, a model-agnostic method that estimates modality reliability and leverages it to guide gradient correction for more coherent backbone optimization, as shown in Fig. 2. Specifically, paired RGB and IR images are fed into separate backbones to extract modality-specific features, which are subsequently fused and passed through a shared detection head to generate the final predictions. During backpropagation, the **r**eliability **e**stimation via **p**arameter **a**ttribution (REPA) module evaluates the reliability of each modality by analyzing the effectiveness and sensitivity of parameters. These reliability scores are then leveraged by the **r**eliability-**g**uided **c**onflict **r**esolution (RGCR) module to guide gradient correction and optimize parameter updates, thereby promoting consistent cross-modal learning and mitigating semantic conflicts.

### 3.1 RELIABILITY ESTIMATION VIA PARAMETER ATTRIBUTION

**Motivation.** Existing work has shown that the features extracted by the model may be unreliable (Laakom et al., 2021; Chen et al., 2023), potentially degrading performance when such features are utilized. Since features are acquired through their corresponding parameters, we propose the **r**eliability **e**stimation via **p**arameter **a**ttribution (REPA) module to assess modality reliability, thereby guiding subsequent gradient correction to learn more reliable features. As shown in the upper-right part of Fig. 2, the REPA module estimates parameter reliability from two complementary aspects: effectiveness and sensitivity (Zhu et al., 2024). Effectiveness quantifies the effect of parameters on prediction performance using counterfactual reasoning, while sensitivity gauges the extent to which the parameter is optimized based on gradient variation. Combining these two measures, we derive a comprehensive reliability score to guide parameter updates.

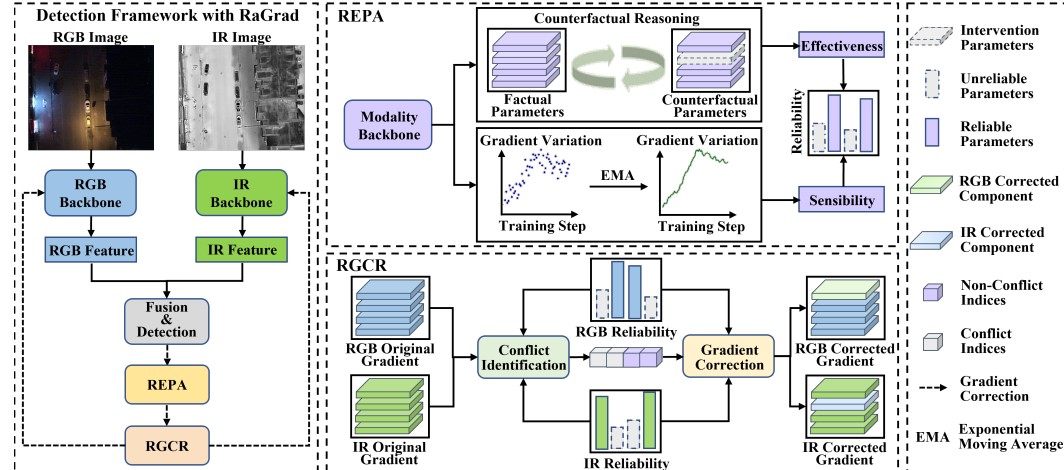

Figure 2: Overview of the proposed RaGrad method. RaGrad consists of two key modules: the **r**eliability **e**stimation via **p**arameter **a**ttribution (REPA) module and the **r**eliability-**g**uided **c**onflict **r**esolution (RGCR) module, which are integrated into existing detection frameworks comprising modality-specific backbones and a fusion module followed by a detection head.

### 3.1.1 EFFECTIVENESS EVALUATION VIA COUNTERFACTUAL REASONING

Given an input $\boldsymbol{x}$ from modality $m \in \{\mathrm{RGB}, \mathrm{IR}\}$ and the complete model parameters $\boldsymbol{\Theta}$, the modality-specific parameter $\boldsymbol{\Theta}_{l,i}^m$ denotes the weights corresponding to the $i$-th output channel of the $l$-th layer in the backbone for modality $m$. We use counterfactual reasoning (Schwab & Karlen, 2019; Li et al., 2025) to evaluate the effectiveness of $\boldsymbol{\Theta}_{l,i}^m$. An intuitive way is first to generate counterfactual parameters $\boldsymbol{\Theta} \backslash \boldsymbol{\Theta}_{l,i}^m$ by intervention that sets $\boldsymbol{\Theta}_{l,i}^m$ to zero (Elsayed & Mahmood, 2024). The effectiveness of $\boldsymbol{\Theta}_{l,i}^m$ is then quantified as the change in loss when the parameter is removed:

$$E_{l,i}^m(\boldsymbol{x}) = \mathcal{L}(\boldsymbol{\Theta} \backslash \boldsymbol{\Theta}_{l,i}^m; \boldsymbol{x}) - \mathcal{L}(\boldsymbol{\Theta}; \boldsymbol{x}), \tag{1}$$

where $\mathcal{L}$ is the loss function. When $E_{l,i}^m(\boldsymbol{x}) > 0$, it represents that removing $\boldsymbol{\Theta}_{l,i}^m$ would incur greater loss, suggesting that $\boldsymbol{\Theta}_{l,i}^m$ has a positive effect on model prediction. Conversely, a negative value implies a negative effect.

However, directly estimating the effect by intervening on each parameter is computationally expensive. To address this, we approximate the counterfactual loss $\mathcal{L}(\boldsymbol{\Theta} \backslash \boldsymbol{\Theta}_{l,i}^m; \boldsymbol{x})$ by performing a second-order Taylor expansion around the current parameter $\boldsymbol{\Theta}_{l,i}^m$ Elsayed & Mahmood (2024), and derive the following quadratic approximation for the effectiveness:

$$\begin{aligned} E_{l,i}^m(\boldsymbol{x}) &= \mathcal{L}(\boldsymbol{\Theta} \backslash \boldsymbol{\Theta}_{l,i}^m, \boldsymbol{x}) - \mathcal{L}(\boldsymbol{\Theta}, \boldsymbol{x}) \\ &\approx \mathcal{L}(\boldsymbol{\Theta}, \boldsymbol{x}) + \frac{\partial \mathcal{L}(\boldsymbol{\Theta}, \boldsymbol{x})}{\partial \boldsymbol{\Theta}_{l,i}^m}(0 - \boldsymbol{\Theta}_{l,i}^m) + \frac{1}{2}\frac{\partial^2 \mathcal{L}(\boldsymbol{\Theta}, \boldsymbol{x})}{\partial(\boldsymbol{\Theta}_{l,i}^m)^2}(0 - \boldsymbol{\Theta}_{l,i}^m)^2 - \mathcal{L}(\boldsymbol{\Theta}, \boldsymbol{x}) \\ &= -\frac{\partial \mathcal{L}(\boldsymbol{\Theta}, \boldsymbol{x})}{\partial \boldsymbol{\Theta}_{l,i}^m}\boldsymbol{\Theta}_{l,i}^m + \frac{1}{2}\frac{\partial^2 \mathcal{L}(\boldsymbol{\Theta}, \boldsymbol{x})}{\partial(\boldsymbol{\Theta}_{l,i}^m)^2}(\boldsymbol{\Theta}_{l,i}^m)^2. \end{aligned} \tag{2}$$

In practice, we employ the AdaHessian method (Yao et al., 2021), which uses the Hutchinson estimator to approximate the Hessian diagonal, reducing the computational complexity from quadratic to linear. This approximation method maintains computational efficiency while preserving the accuracy of the estimation, thus enabling scalable and accurate effectiveness assessment during training.

Since raw effectiveness scores may vary significantly across parameters and modalities, we apply z-score normalization followed by a sigmoid function to stabilize the values and enable consistent comparison. The normalized effectiveness score $\hat{E}_{l,i}^m$ is defined as:

$$\hat{E}_{l,i}^m = \sigma\left(\frac{E_{l,i}^m - \mu_e}{s_e + \epsilon}\right), \tag{3}$$

where $\mu_e$ and $s_e$ denote the mean and standard deviation of effectiveness scores computed across both modalities, $\epsilon$ is a small constant for numerical stability, and $\sigma(\cdot)$ is the sigmoid function. The normalized score $\hat{E}_{l,i}^m \in (0, 1)$ is a scalar, with higher values indicating a greater contribution of the corresponding parameter to the model's prediction.

### 3.1.2 SENSITIVITY EVALUATION VIA GRADIENT VARIATION

During training, different parameters may receive varying levels of optimization attention (Frankle & Carbin, 2019). To identify actively optimized parameters, we define a sensitivity score that reflects optimization intensity based on temporal gradient variation. A higher score indicates greater variation, suggesting that the model more actively optimizes the corresponding parameter (Wang et al., 2024b). Specifically, we compute this score as the exponential moving average of squared deviations between the current gradient and its historical mean, effectively combining past and current gradient information to yield a more robust estimate against noise (Tarvainen & Valpola, 2017). At training step $t$, the gradient of the parameter subset $\Theta_{l,i}^m$ is denoted as $\nabla\Theta_{l,i}^{m,t}$, with $\mu_{l,i}^{m,t}$ representing its historical mean. The sensitivity $S_{l,i}^{m,t}$ is defined as:

$$S_{l,i}^{m,t} = \alpha S_{l,i}^{m,t-1} + (1 - \alpha)(\nabla\Theta_{l,i}^{m,t} - \mu_{l,i}^{m,t})^2, \tag{4}$$

where the decay factor $\alpha$ balances the influence of current and historical gradient information. The sensitivity score $S_{l,i}^{m,t}$ is normalized following the same procedure as in Eq. 3, yielding the final scalar score $\hat{S}_{l,i}^m \in (0, 1)$. Higher values suggest that the model places greater optimization focus on the corresponding parameter during training.

After normalizing the effectiveness and sensitivity scores, we compute the final reliability score as:

$$R_{l,i}^m = \hat{E}_{l,i}^m \cdot \hat{S}_{l,i}^m. \tag{5}$$

The score $R_{l,i}^m \in (0, 1)$ reflects the reliability of the corresponding modality-specific parameter, where higher values indicate more reliable feature extraction. This score subsequently serves as a guiding signal for gradient correction.

## 3.2 RELIABILITY-GUIDED CONFLICT RESOLUTION

**Motivation.** Semantic conflicts occur when the same object is represented inconsistently across modalities, often leading to gradient conflicts (Wu et al., 2022). These conflicts can disrupt coherent parameter updates and hinder the learning of reliable features (Zhang et al., 2023a). To address this, we propose the **r**eliability-**g**uided **c**onflict **r**esolution (RGCR) module, which leverages modality reliability to guide gradient correction. As shown in the lower-right part of Fig. 2, RGCR first identifies conflicts using a dual-criteria strategy that jointly considers gradient direction and modality reliability. It then resolves these conflicts via reliability-guided gradient correction, refining the gradients of less reliable modalities using signals from more reliable ones. This promotes more consistent feature learning and mitigates cross-modal semantic conflicts.

### 3.2.1 DUAL-CRITERIA CONFLICT IDENTIFICATION

Effective gradient correction relies on accurately detecting conflicts during training. Conventional methods (Wang et al., 2024a) typically define conflicts as any pair of gradients with negative cosine similarity and adjust them accordingly. However, such a criterion may lead to unnecessary adjustments when gradient directions deviate only slightly from the threshold (Liu et al., 2021a).

To overcome the above limitations, we introduce a dual-criteria conflict identification strategy that jointly considers gradient direction inconsistency and reliability gap across modalities. Specifically, a directional inconsistency is detected when the cosine similarity between gradients of corresponding RGB and IR backbone parameters falls below the mean of negative similarities in the layer. The reliability gap is considered significant if it exceeds the layer-wise average, indicating that one modality is more reliable than the other. A conflict is identified only when both conditions are simultaneously satisfied. Formally, the conflict set $\mathcal{C}$ is defined as the index pairs $(l, i)$ over all backbone

parameter positions that meet both criteria:

$$\mathcal{C} = \left\{ (l,i) \mid (\cos_{l,i} < \mu_l^{\cos}) \wedge (\left| R_{l,i}^{\text{RGB}} - R_{l,i}^{\text{IR}} \right| > \mu_l^R) \right\}, \tag{6}$$

where $\cos_{l,i}$ is the cosine similarity between the gradients of $\Theta_{l,i}^{\text{RGB}}$ and $\Theta_{l,i}^{\text{IR}}$. $\mu_l^{\cos}$ and $\mu_l^R$ denote the average negative cosine similarity and reliability gap across all output channels in the $l$-th layer, respectively. By jointly considering directional inconsistency and reliability disparity, it establishes a robust and discerning criterion for conflict identification.

### 3.2.2 RELIABILITY-GUIDED GRADIENT CORRECTION

After identifying the conflicting set $\mathcal{C}$, we determine the more reliable modality $\mathcal{R}$ for each $(l,i) \in \mathcal{C}$ based on its reliability score:

$$\mathcal{R} = \arg \max_{m \in \{\text{RGB,IR}\}} R_{l,i}^m. \tag{7}$$

Accordingly, the less reliable modality is denoted as $\bar{m}$. To mitigate the adverse impact of less reliable gradients, we perform orthogonal decomposition on the gradient from $\bar{m}$, removing its projection onto the more reliable gradient and retaining only the informative components in the orthogonal direction. Formally, given the gradients of the unreliable and reliable modalities, denoted as $g_{l,i}^{\bar{m}}$ and $g_{l,i}^{\mathcal{R}}$ respectively. The orthogonal component of $g_{l,i}^{\bar{m}}$ relative to $g_{l,i}^{\mathcal{R}}$ is computed as:

$$\hat{g}_{l,i}^{\bar{m}} = g_{l,i}^{\bar{m}} - \frac{g_{l,i}^{\bar{m}} \cdot g_{l,i}^{\mathcal{R}}}{\|g_{l,i}^{\mathcal{R}}\|_2^2 + \epsilon} \cdot g_{l,i}^{\mathcal{R}}, \tag{8}$$

where $\epsilon$ is the same stability constant as in Eq. 3. We then refine the gradient from the less reliable modality through reliability-guided blending, which combines it with the gradient from the more reliable one, weighted by their respective reliability, thereby promoting more consistent optimization. The final corrected gradient $\tilde{g}_{l,i}^{\bar{m}}$ is obtained as:

$$\tilde{g}_{l,i}^{\bar{m}} = (1 - \lambda_{l,i})\hat{g}_{l,i}^{\bar{m}} + \lambda_{l,i} g_{l,i}^{\mathcal{R}}, \tag{9}$$

where the weighting factor $\lambda_{l,i} \in [0,1]$ controls the contribution of the more reliable modality in blending, with larger values indicating stronger guidance, and is given by:

$$\lambda_{l,i} = \sigma \left( \frac{\left| R_{l,i}^{\text{RGB}} - R_{l,i}^{\text{IR}} \right|}{R_{l,i}^{\text{RGB}} + R_{l,i}^{\text{IR}} + \epsilon} \right), \tag{10}$$

where $\sigma(\cdot)$ denotes the sigmoid function. Finally, the corrected gradients $\tilde{g}_{l,i}^{\bar{m}}$ and the reliable modality gradients $g_{l,i}^{\mathcal{R}}$ are used to update the model parameters, promoting the learning of more reliable feature and alleviating cross-modal semantic conflicts.

To better illustrate the above procedure, Fig. 3 illustrates the reliability-guided gradient correction after conflict identification. First, we perform orthogonal decomposition on the unreliable modality gradient $g_{l,i}^{\bar{m}}$, removing its conflicting component and obtaining the orthogonal component $\hat{g}_{l,i}^{\bar{m}}$ (Eq. 8). Then, we apply reliability-guided blending, combining $\hat{g}_{l,i}^{\bar{m}}$ and the reliable modality gradient $g_{l,i}^{\mathcal{R}}$ based on their reliability, producing $g_{l,i}^{ur}$ and $g_{l,i}^{r}$ as their respective weighted components, which correspond to the two terms in Eq. 9 and jointly form the

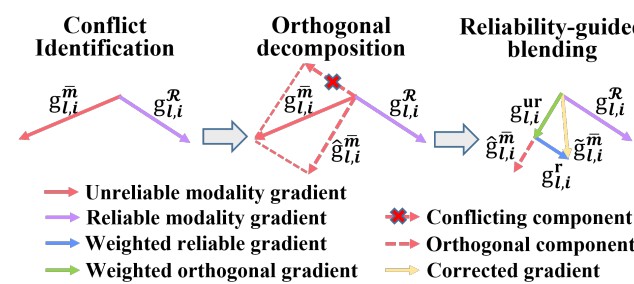

Figure 3: Illustration of the reliability-guided gradient correction after conflict identification. It involves two steps: orthogonal decomposition to remove the conflicting component from the unreliable modality gradient, and reliability-guided blending to combine the orthogonal component with the reliable modality gradient, weighted by their reliability.

Table 1: Performance comparison on VEDAI.

| Detector | Method | mAP$_{50}$ | mAP$_{75}$ | mAP |
|---|---|---|---|---|
| S$^2$A-Net | S$^2$A-Net | 73.0 | 29.5 | 35.4 |
| | S$^2$A-Net + Ours | **73.6** | **29.9** | **37.3** |
| | CSSA | 74.0 | 31.7 | 38.1 |
| | CSSA + Ours | **74.5** | **32.8** | **38.7** |
| | C$^2$Former | 78.1 | 32.3 | 39.1 |
| | C$^2$Former + Ours | **79.7** | **33.4** | **40.3** |
| YOLOv5 (OBB) | CALNet | 84.3 | 56.8 | 52.1 |
| | CALNet + Ours | **86.5** | **57.3** | **53.8** |
| | CDC-YF | 82.2 | 54.4 | 48.3 |
| | CDC-YF + Ours | **84.3** | **56.7** | **50.7** |
| | EI$^2$Det | 86.5 | 62.8 | 54.0 |
| | EI$^2$Det + Ours | **87.8** | **64.6** | **55.3** |

Table 2: Performance comparison on LLVIP.

| Detector | Method | AP$_{50}$ | AP$_{75}$ | mAP |
|---|---|---|---|---|
| Faster R-CNN | Faster R-CNN | 93.1 | 63.5 | 56.8 |
| | Faster R-CNN + Ours | **93.5** | **64.3** | **57.8** |
| | CSSA | 94.4 | 66.3 | 59.0 |
| | CSSA + Ours | **95.1** | **66.9** | **59.9** |
| | C$^2$Former | 95.7 | 71.2 | 61.3 |
| | C$^2$Former + Ours | **95.9** | **72.4** | **62.1** |
| YOLOv5 | CALNet | 97.1 | 76.2 | 66.1 |
| | CALNet + Ours | **97.3** | **76.9** | **67.2** |
| | CDC-YF | 96.4 | 75.0 | 64.1 |
| | CDC-YF + Ours | **96.5** | **75.3** | **64.8** |
| | EI$^2$Det | 97.0 | 77.1 | 66.7 |
| | EI$^2$Det + Ours | **97.3** | **78.9** | **67.5** |

corrected gradient $\tilde{g}_{l,i}^{\bar{m}}$. Reliability-guided gradient correction ensures consistent parameter updates across modalities, promotes the learning of more reliable features, and mitigates semantic conflicts. The detailed algorithm and analysis of the time and space complexity for our proposed RaGrad are provided in Appendix A.

# 4 EXPERIMENTS

## 4.1 EXPERIMENTAL SETTINGS

**Datasets.** Following existing works (Wang et al., 2024c; Shang et al., 2025), we conduct experiments on three widely used visible-infrared object detection datasets. **VEDAI** (Razakarivony & Jurie, 2016) is an aerial-view vehicle dataset consisting of 1,210 visible-infrared image pairs at two resolutions. Following prior works (Zhang et al., 2023b; 2024), we adopt the higher-resolution version and the eight-category setting, with 1,089 pairs for training and 121 for testing. **LLVIP** (Jia et al., 2021) comprises 15,488 well-aligned pedestrian image pairs, tailored for low-light scenarios and split into 12,025 for training and 3,463 for testing. **DroneVehicle** (Sun et al., 2022) is a drone-captured dataset containing 28,439 visible-infrared image pairs across five vehicle categories, split into 17,990 for training, 1,469 for validation, and 8,980 for testing.

**Evaluation Metrics.** Following prior works (Hu et al., 2025), we adopt Average Precision (AP) as the evaluation metric, which considers a prediction correct if its Intersection over Union (IoU) with the ground-truth box exceeds a specified threshold. We report AP at IoU thresholds of 0.50 and 0.75 (denoted as AP$_{50}$ and AP$_{75}$), and compute mAP$_{50}$ and mAP$_{75}$ as the mean AP across all categories. In addition, we report mAP averaged over IoU thresholds from 0.50 to 0.95 with a step size of 0.05, providing a more comprehensive evaluation. Higher mAP indicates better performance. All reported results are averaged over five runs with different random seeds.

**Implementation Details.** We compare our method with 7 baselines, including the extended dual-modality S$^2$ANet (Han et al., 2022) and Faster R-CNN (Ren et al., 2015), as well as 5 visible-infrared object detection methods: CSSA (Cao et al., 2023), CALNet (He et al., 2023), CDC-YOLOFusion (abbreviated as CDC-YF) (Wang et al., 2024c), C$^2$Former (Yuan & Wei, 2024), and EI$^2$Det (Hu et al., 2025). These methods are built upon 4 detectors: S$^2$ANet and YOLOv5(OBB) (Yang & Yan, 2022) for oriented detection; Faster R-CNN and YOLOv5 (Jocher, 2020) for horizontal detection. For fair comparison, we use the official implementations of all baselines and integrate our method as a plug-in module with their default hyperparameter settings. The sensitivity decay factor $\alpha$ in REPA is set to 0.95. All experiments use a single NVIDIA Tesla V100 GPU. More details are provided in Appendix C.

## 4.2 MAIN RESULTS

**Results on VEDAI Dataset.** As shown in Table 1, our method consistently improves detection performance across all six baselines on the VEDAI dataset. The most significant gains appear on CALNet and CDC-YF. Specifically, integrating our approach into CALNet boosts mAP$_{50}$ from

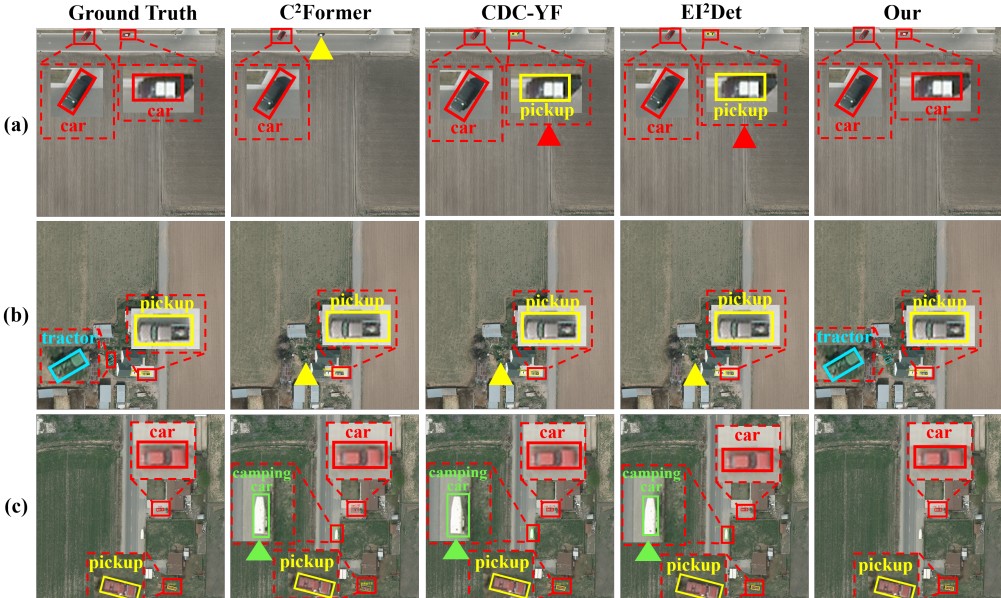

Figure 4: Visual comparison of detection results on the VEDAI dataset. Red dashed boxes highlight zoomed-in regions for clarity. Triangles denote typical errors: red for misclassifications, yellow for missed detection, and green for false positives.

84.3% to 86.5%, $mAP_{75}$ from 56.8% to 57.3%, and overall mAP from 52.1% to 53.8%. For CDC-YF, our method raises the scores from 82.2%, 54.4%, and 48.3% to 84.3%, 56.7%, and 50.7%, respectively. Notably, $EI^2Det$ with our method achieves the best overall performance, reaching 87.8% $mAP_{50}$, 64.6% $mAP_{75}$, and 55.3% mAP. Across all baselines, our approach achieves average improvements of 1.4% in $mAP_{50}$, 1.2% in $mAP_{75}$, and 1.5% in mAP, demonstrating its effectiveness in enhancing detection accuracy under aerial-view conditions.

**Results on LLVIP Dataset.** Table 2 summarizes the results on the LLVIP dataset, designed for pedestrian detection under extreme low-light conditions. Our method achieves consistent gains across all baselines, particularly in $AP_{75}$ and mAP, indicating enhanced fine-grained localization. When combined with $EI^2Det$, our method achieves the highest overall accuracy, improving $AP_{50}$ from 97.0% to 97.3%, $AP_{75}$ from 77.1% to 78.9%, and mAP from 66.7% to 67.5%. On average, it achieves improvements of 0.3% in $AP_{50}$ and 0.9% in both $AP_{75}$ and mAP, highlighting the robustness of our method in low-illumination scenarios. Further results on the DroneVehicle dataset are shown in Appendix D.

### 4.3 ABLATION STUDY

We conduct ablation studies on the VEDAI dataset with $EI^2Det$ as baseline to assess the contributions of effectiveness and sensitivity components in the REPA module. As shown in Table 3, the baseline without either component achieves 86.5% $mAP_{50}$, 62.8% $mAP_{75}$, and 54.0% mAP. Activating only the sensitivity component slightly degrades performance, likely because it captures update intensity without assessing actual contribution to prediction, leading to suboptimal correction. In contrast, enabling only the effectiveness component improves all metrics, highlighting its ability to identify effective parameters. The full REPA module achieves the best performance with 87.8% $mAP_{50}$, 64.6% $mAP_{75}$, and 55.3% mAP, confirming their complementary benefits in enhancing detection

Table 3: Ablation study on the effectiveness and sensitivity components of the REPA module on the VEDAI dataset.

| Effectiveness | Sensitivity | $mAP_{50}$ | $mAP_{75}$ | mAP |
|:---:|:---:|:---:|:---:|:---:|
| × | × | 86.5 | 62.8 | 54.0 |
| × | ✓ | 85.8 | 62.1 | 53.6 |
| ✓ | × | 87.4 | 63.9 | 54.8 |
| ✓ | ✓ | **87.8** | **64.6** | **55.3** |

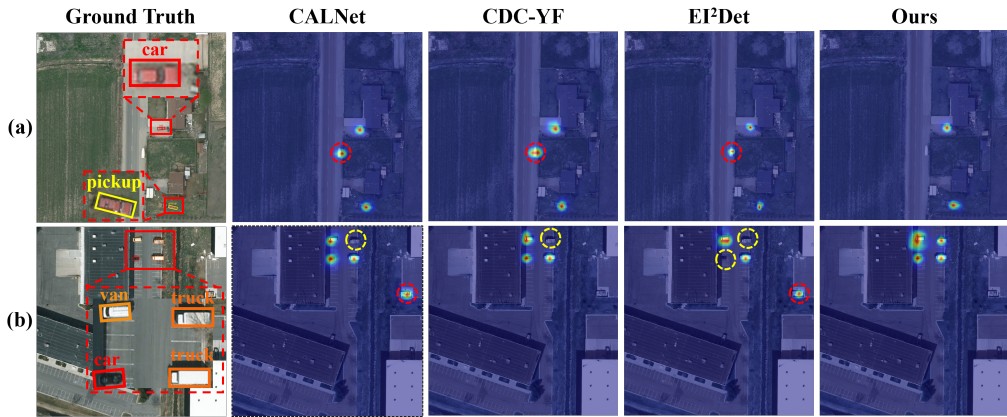

Figure 5: Visual comparisons of saliency maps on the VEDAI dataset. Brighter regions indicate stronger model attention. Red dashed circles mark attention to irrelevant regions, and yellow dashed circles indicate overlooked target regions.

performance. In addition, we evaluate the impact of the hyperparameter $\alpha$ in the sensitivity computation (defined in Eq. 4) and analyze the contribution of each component in the RGCR module. Detailed results are provided in Appendix E.

### 4.4 VISUALIZATION RESULTS

**Visualization of Detection Results.** Fig. 4 presents qualitative comparisons of detection results. Each row depicts a distinct scene: the first column displays the ground truth annotations, and the remaining columns present results from $C^2$Former, CDC-YF, EI$^2$Det, and our method. As illustrated, the baselines exhibit common errors, including misclassifications, missed detections, and false positives. Specifically, in row (a), $C^2$Former completely misses the "car" (yellow triangles), while CDC-YF and EI$^2$Det misclassify it as a "pickup" (red triangles); in row (b), all baselines miss the "tractor"; and in row (c), they produce false positives on the "camping car" (green triangles). In contrast, our method accurately recalls missed objects, corrects misclassifications, and suppresses false positives, resulting in more precise and complete detections in challenging aerial scenarios.

**Visualization of Saliency Maps.** To further analyze how different methods allocate attention to target regions, we visualize their fused saliency maps in Fig. 5 using the LayerCAM (Jiang et al., 2021). Each row represents a different scene, where the first column shows the ground truth annotations and the remaining columns display the saliency maps of CALNet, CDC-YF, EI$^2$Det, and our method, respectively. As observed, the baseline methods may attend to irrelevant areas (red dashed circles) or overlook critical target regions (yellow dashed circles). Specifically, in row (a), all baselines incorrectly focus on non-target areas, increasing the risk of false positives; in row (b), all fail to attend to the "tractor", which may result in missed detection. In contrast, our method suppresses irrelevant distractions and accurately attends to target regions, thereby reducing both false positives and missed detections. These results confirm that our approach facilitates the learning of more reliable features, enhancing detection performance in complex scenarios.

## 5 CONCLUSION

In this paper, we propose RaGrad, a model-agnostic approach that mitigates cross-modal semantic conflicts in visible–infrared object detection via reliability-guided gradient correction. Specifically, we first propose the **r**eliability **e**stimation via **p**arameter **a**ttribution (REPA) module, which evaluates modality reliability by assessing parameter effectiveness via counterfactual reasoning and sensitivity via gradient variation. Second, we introduce the **r**eliability-**g**uided **c**onflict **r**esolution (RGCR) module, which leverages reliability scores to optimize parameter updates of less reliable modalities via gradient correction, thereby improving cross-modal consistency and detection performance. Extensive experiments across multiple detection frameworks on three challenging datasets demonstrate the superiority and generalizability of our method.

ETHICS STATEMENT

This work adheres to the ICLR Code of Ethics. No human or animal subjects were involved in the research, and all data used were sourced from publicly available datasets that comply with privacy protection standards. We ensured that all relevant usage policies were followed, minimizing any potential ethical concerns. Therefore, there are no ethical issues associated with this work.

REPRODUCIBILITY STATEMENT

We have taken steps to ensure that the results presented in this paper are reproducible. Detailed information about the experimental setup, including training configurations and hardware details, can be found in the 4.1 section and Appendix C. Relevant code is provided as supplementary materials to facilitate replication of our experiments.

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

## A  ALGORITHM AND COMPLEXITY ANALYSIS OF RAGRAD

Algorithm 1 describes the RaGrad method for visible-infrared object detection. Given a training dataset $\mathcal{D}$ consisting of paired visible and infrared images with annotations, and the initial model parameters $\Theta$, the training proceeds for $e$ epochs. In each epoch, for every mini-batch $x \subset \mathcal{D}$, the task loss $\mathcal{L}(\Theta, x)$ is computed according to the original baseline detector. RaGrad then estimates the parameter-level effectiveness $\hat{E}^m$ and sensitivity $\hat{S}^m$ for each modality (Eq. 2–4). These are further combined to derive the reliability score $R^m$ (Eq. 5). Based on the computed reliability scores and the gradient direction, a conflict set $\mathcal{C}$ is identified (Eq. 6). For each conflict, the more reliable modality is determined

---

**Algorithm 1: RaGrad for Visible-Infrared Object Detection**

**Input:** dataset $\mathcal{D}$, model parameters $\Theta$, number of epochs $e$
**Output:** Updated model parameters $\Theta$

1: **for** epoch = 1 to $e$ **do**
2:     **for** each mini-batch $x \subset \mathcal{D}$ **do**
3:         Compute loss $\mathcal{L}(\Theta, x)$
4:         Estimate effectiveness $\hat{E}^m$ by Eq. 2–3
5:         Estimate sensitivity $\hat{S}^m$ by Eq. 3–4
6:         Compute reliability $R^m$ by Eq. 5
7:         Identify conflict set $\mathcal{C}$ by Eq. 6
8:         **for** each $(l, i) \in \mathcal{C}$ **do**
9:             Determine reliable modality $\mathcal{R}$ by Eq. 7
10:        Perform gradient correction by Eq. 8–10
11:         **end for**
12:         Update $\Theta$ using corrected gradients
13:     **end for**
14: **end for**

---

(Eq. 7), and the gradient from the less reliable modality is corrected via orthogonal projection and reliability-guided blending (Eq. 8–10). After all corrections, the model parameters $\Theta$ are updated using the corrected gradients, and the final output is $\Theta$ after $e$ training epochs. This procedure promotes the learning of more reliable features and enhances cross-modal consistency.

RaGrad introduces only minimal computational overhead compared to standard backpropagation during training. For each baseline method, standard backpropagation has both time and space complexity of $\mathcal{O}(P)$, where $P$ denotes the number of model parameters. When integrated into these methods, RaGrad performs several lightweight operations. First, the effectiveness score $\hat{E}^m$ is approximated using a second-order Taylor expansion, where the diagonal approximation of the Hessian enables efficient computation in linear time, incurring a cost of $\mathcal{O}(P)$. Second, identifying conflicts requires scanning all parameters, contributing an additional $\mathcal{O}(P)$ cost. Finally, gradient correction over the conflict set $\mathcal{C}$ introduces an overhead of $\mathcal{O}(|\mathcal{C}|)$, where $|\mathcal{C}| \ll P$. As a result, RaGrad adds an extra time complexity of $\mathcal{O}(P + P + |\mathcal{C}|) \approx \mathcal{O}(P)$, which remains linear and comparable to standard backpropagation. In terms of space, RaGrad maintains only a few scalar statistics per parameter (e.g., effectiveness, sensitivity, and reliability), resulting in an additional space complexity of $\mathcal{O}(P)$. Therefore, both the time and space overhead introduced by RaGrad are negligible, and the overall computational complexity remains linear in the number of parameters, consistent with that of each baseline method. Importantly, RaGrad introduces no additional overhead during inference, and the inference speed remains identical to that of the baseline methods.

Table 4 further validates the theoretical complexity analysis by comparing the training efficiency and inference speed of baseline EI$^2$Det and our method integrated into it. Both methods are evaluated on the VEDAI dataset with a batch size of 8 and an input resolution of 1024. RaGrad does not increase the number of model parameters, remaining identical to the baseline. In terms of memory usage during training, RaGrad incurs only a marginal increase in peak memory, rising from 21.60 GB for the baseline to 21.85 GB. Similarly, RaGrad introduces slight overhead in training throughput and latency, processing 11.71 images per second with an iteration time of 683.43 ms, while the baseline achieves 12.10 images per second with 661.40 ms per iteration. Importantly, the inference speed of RaGrad remains virtually unchanged, reaching 16.05 FPS compared to 16.06 FPS for the baseline. These results demonstrate that RaGrad introduces only negligible overhead during training and imposes no additional cost during inference, fully aligning with the theoretical analysis.

## B  COMPARISON BASELINES

We provide a brief overview of each baseline method used in our experiments, including the extended dual-modality S$^2$ANet (Han et al., 2022) and Faster R-CNN (Ren et al., 2015), as well as

Table 4: Comparison of training efficiency and inference speed across different methods.

| Method | Training | | | | Inference |
|---|---|---|---|---|---|
| | Params (M)↓ | Peak mem (GB)↓ | Throughput (img/s)↑ | Latency (ms/iter)↓ | FPS (img/s)↑ |
| EI$^2$Det | **129.59** | **21.60** | **12.10** | **661.40** | **16.06** |
| EI$^2$Det+Ours | **129.59** | 21.85 | 11.71 | 683.43 | 16.05 |

Table 5: Training settings for all baseline methods. "–" denotes not applicable. "*" indicates settings that vary across datasets.

| Method | Optimizer | Learning rate | Momentum | Weight decay | Epoch* | Batch size* |
|---|---|---|---|---|---|---|
| Faster R-CNN | SGD | 0.0025 | 0.9 | 0.0001 | 24 | 8 |
| S$^2$ANet | SGD | 0.01 | 0.9 | 0.0001 | 24 | 8 |
| CSSA | AdamW | 0.00025 | – | 0.0001 | 10 | 16 |
| CALNet | SGD | 0.035 | 0.85 | 0.001 | 50 | 4 |
| CDC-YF | SGD | 0.001 | 0.937 | 0.0005 | 500 | 8 |
| C$^2$Former | SGD | 0.001 | 0.9 | 0.0001 | 24 | 2 |
| EI$^2$Det | SGD | 0.01 | 0.8 | 0.001 | 120 | 32 |

5 representative visible-infrared object detection methods: CSSA (Cao et al., 2023), CALNet (He et al., 2023), CDC-YOLOFusion (abbreviated as CDC-YF) (Wang et al., 2024c), C$^2$Former (Yuan & Wei, 2024), and EI$^2$Det (Hu et al., 2025). These methods are built upon 4 widely used detectors: S$^2$ANet and YOLOv5(OBB) (Yang & Yan, 2022) for oriented detection; Faster R-CNN and YOLOv5 (Jocher, 2020) for horizontal detection.

- The extended dual-modality versions of S$^2$ANet and Faster R-CNN first extract features from RGB and IR modalities using separate backbones, then fuse them via simple element-wise addition, and finally pass the fused features to the detection heads for prediction.

- CSSA introduces a lightweight fusion module that integrates RGB and IR features through a combination of channel switching and spatial attention mechanisms, aiming to balance detection performance and computational efficiency in multimodal object detection.

- CALNet tackles the challenge of semantic conflicts in multispectral object detection by first rectifying inconsistent modality features based on contextual similarity, then selectively fusing semantically coherent and complementary features across modalities.

- CDC-YF introduces a cross-scale dynamic fusion module to adaptively extract and integrate RGB and IR features based on data distribution. It further enhances cross-modal representation by leveraging a data swapping strategy, disparity-aware attention, and a kernel interaction loss tailored for bimodal feature learning.

- C$^2$Former introduces a calibrated and complementary transformer architecture that aligns and fuses RGB-IR features using inter-modality cross-attention, while reducing computational cost via adaptive feature sampling.

- EI$^2$Det addresses the challenge of balancing the contributions of RGB and IR information under varying lighting conditions by introducing an illumination-aware detector that adaptively fuses multimodal features based on scene illumination and leverages edge information to enhance boundary localization.

## C   IMPLEMENTATION DETAILS

Since our method can be seamlessly integrated into most existing detection frameworks, we adopt the original training configurations of each baseline to evaluate the performance improvements brought by our method. For clarity and reproducibility, we summarize the complete training configurations for all baseline methods in Table 5. These settings, including the optimizer type, learning

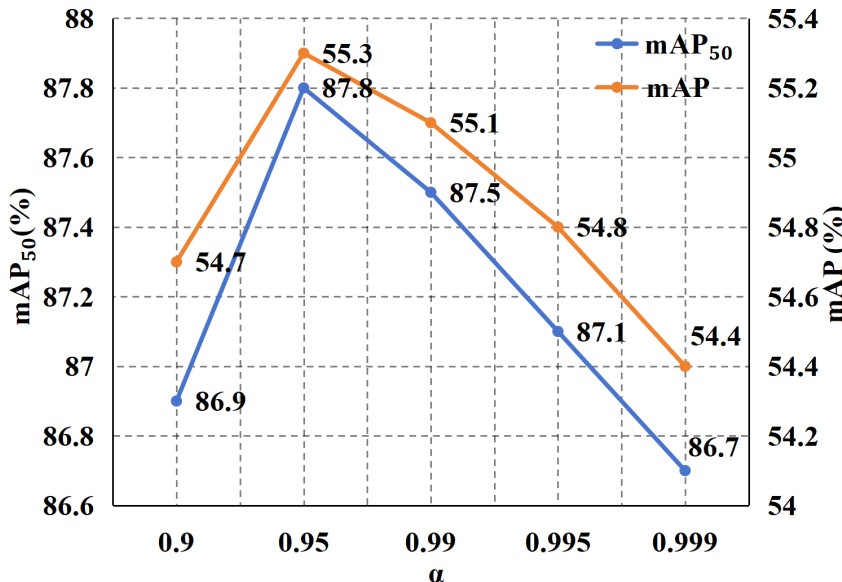

Figure 6: Performance variation with different values of $\alpha$ in the sensitivity computation.

rate, momentum, and weight decay, strictly follow the original implementations provided in the respective papers. Notably, the number of training epochs and batch sizes may vary slightly across datasets. For the LLVIP and DroneVehicle datasets, we follow the settings listed in Table 5, which are consistent with the original papers. For the VEDAI dataset, due to its relatively small scale and the lack of reported configurations in most baseline papers, we follow existing works (Zhang et al., 2023b; Wang et al., 2024c) and adopt a unified setup of 300 epochs and a batch size of 4 for all methods to ensure consistent comparison.

# D  RESULTS ON DRONEVEHICLE DATASET

Table 6 presents the evaluation results on the DroneVehicle dataset, comprising large-scale aerial images with complex backgrounds and varying illumination. Our method consistently enhances the performance of all six baselines. Notably, EI$^2$Det achieves the highest overall accuracy when integrated with our method, reaching 79.0% mAP$_{50}$, 68.7% mAP$_{75}$, and 57.6% mAP. On average, our method contributes a 0.5% gain in each metric, underscoring its effectiveness and robustness in handling challenging aerial scenarios with diverse illumination conditions.

Table 6: Performance comparison on DroneVehicle.

| Detector | Method | mAP$_{50}$ | mAP$_{75}$ | mAP |
|---|---|---|---|---|
| S$^2$A-Net | S$^2$A-Net | 71.4 | 45.8 | 42.7 |
| | S$^2$A-Net + Ours | **71.9** | **46.1** | **43.0** |
| | CSSA | 71.8 | 46.4 | 43.2 |
| | CSSA + Ours | **72.1** | **47.2** | **43.4** |
| | C$^2$Former | 74.3 | 50.0 | 45.1 |
| | C$^2$Former + Ours | **74.8** | **50.3** | **45.6** |
| YOLOv5 (OBB) | CALNet | 75.5 | 62.6 | 53.9 |
| | CALNet + Ours | **76.2** | **63.2** | **54.5** |
| | CDC-YF | 76.4 | 63.9 | 55.1 |
| | CDC-YF + Ours | **77.0** | **64.5** | **55.5** |
| | EI$^2$Det | 78.3 | 67.5 | 57.0 |
| | EI$^2$Det + Ours | **79.0** | **68.7** | **57.6** |

# E  ABLATION STUDY

**RGCR Module Components.** To assess the contributions of the dual-criteria conflict identification (DCCI) and reliability-guided gradient correction (RGGC) in the RGCR module, we conduct an ablation study using the EI$^2$Det model as the baseline on the VEDAI dataset. As shown in Table 7, when neither component is applied, the method relies solely on cosine similarity for conflict identification and orthogonal decomposition for conflict resolution, resulting in 85.4 mAP$_{50}$, 60.8 mAP$_{75}$,

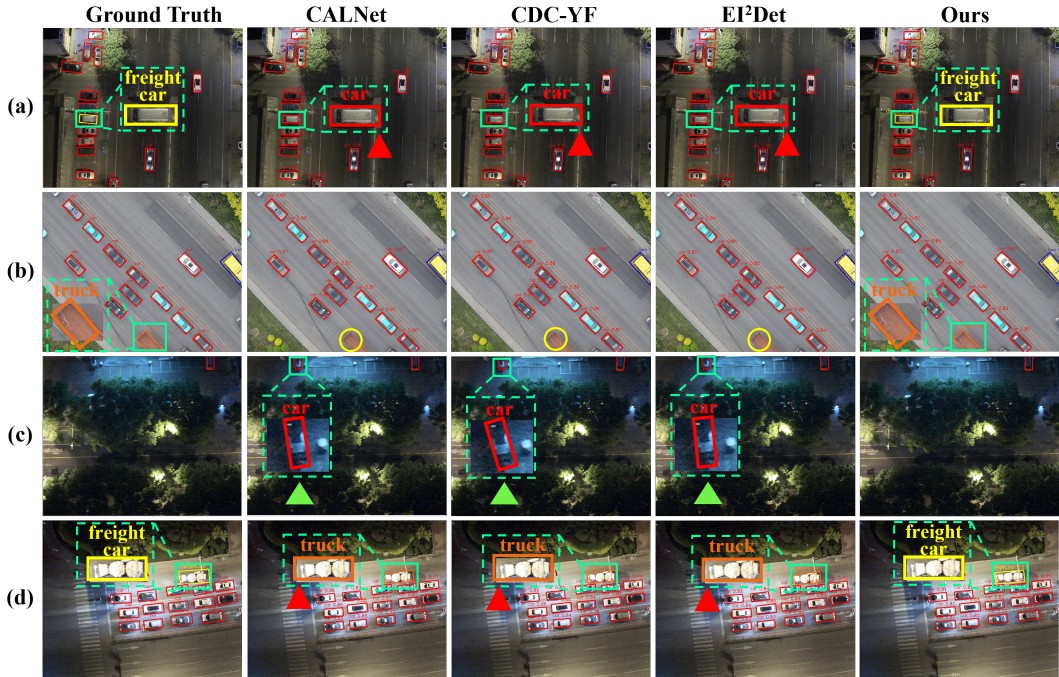

Figure 7: Visual comparison of detection results on the DroneVehicle dataset. Green dashed boxes mark zoomed-in regions. Red triangles indicate misclassifications, green triangles denote false positives, and yellow circles highlight missed detections.

Table 7: Ablation study of the dual-criteria conflict identification (DCCI) and reliability-guided gradient correction (RGGC) components of the RGCR module on the VEDAI dataset.

| DCCI | RGGC | $mAP_{50}$ | $mAP_{75}$ | mAP |
|---|---|---|---|---|
| × | × | 85.4 | 60.8 | 53.1 |
| × | ✓ | 87.3 | 63.5 | 54.3 |
| ✓ | × | 87.2 | 64.3 | 55.0 |
| ✓ | ✓ | **87.8** | **64.6** | **55.3** |

Table 8: Performance comparison with other conflict resolution methods the VEDAI Dataset. "*" indicates that the conflict resolution method is applied to visible-infrared object detection.

| Method | $mAP_{50}$ | $mAP_{75}$ | mAP |
|---|---|---|---|
| PCGrad* | 84.9 | 61.8 | 52.8 |
| IMTL* | 86.4 | 63.4 | 54.2 |
| Ours | **87.8** | **64.6** | **55.3** |

and 53.1 mAP. Activating only RGGC improves the performance to 87.3 $mAP_{50}$, 63.5 $mAP_{75}$, and 54.3 mAP. When DCCI is enabled alone, the model achieves 87.2 $mAP_{50}$, 64.3 $mAP_{75}$, and 55.0 mAP. Finally, combining both components yields the best performance, with 87.8 $mAP_{50}$, 64.6 $mAP_{75}$, and 55.3 mAP. These results highlight the complementary contributions of both components in mitigating semantic conflicts and enhancing detection performance.

To further validate the effectiveness of the RGCR module, we compare it with other conflict resolution methods, such as PCGrad (Yu et al., 2020) and IMTL (Liu et al., 2021b), on the VEDAI dataset using EI$^2$Det as the baseline. As shown in Table 8, our method consistently outperforms both PCGrad and IMTL, highlighting the superiority of our gradient correction approach in resolving conflicts and enhancing detection accuracy.

**Hyperparameter $\alpha$.** We conduct an ablation study on the VEDAI dataset using EI$^2$Det as the baseline to examine the effect of the hyperparameter $\alpha$ in the sensitivity computation (defined in Eq. 4). Following prior works (Grill et al., 2020), we vary $\alpha$ across a range from 0.9 to 0.999 to investigate its impact on detection performance, as illustrated in Fig. 6. The results show that setting $\alpha$ to 0.95 yields the highest performance, achieving 87.8% $mAP_{50}$ and 55.3% mAP. When $\alpha$ is either too small or too large, the model's performance deteriorates. This trend suggests that an excessively

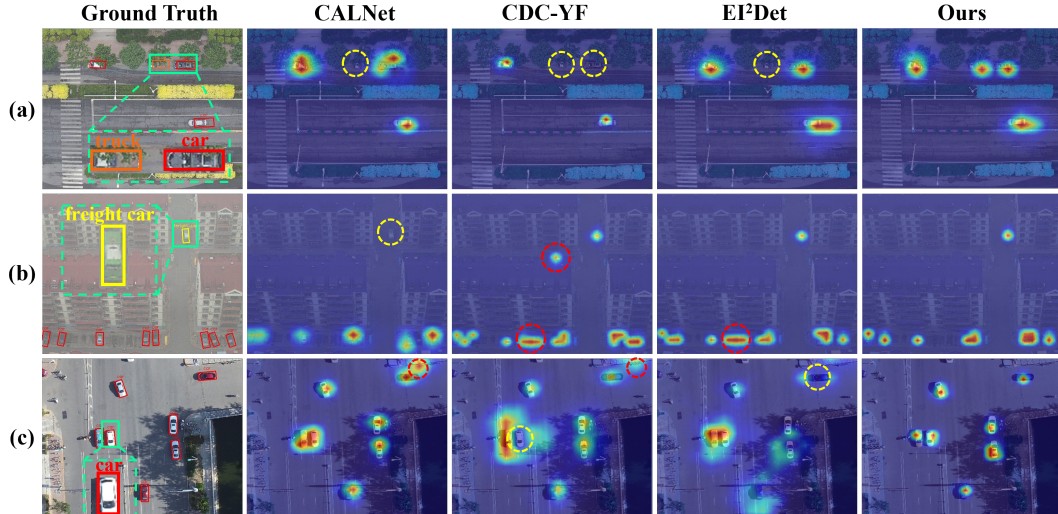

Figure 8: Visual comparisons of saliency maps on the DroneVehicle dataset. Brighter regions indicate stronger model attention. Red dashed circles mark attention to irrelevant regions, and yellow dashed circles indicate overlooked target regions.

low $\alpha$ may amplify short-term fluctuations in gradient dynamics, leading to unstable sensitivity estimates, while an overly high $\alpha$ may over-smooth the temporal gradient variations, masking the true optimization intensity of individual parameters. Therefore, a moderate $\alpha$ around 0.95 provides a favorable trade-off between stability and responsiveness, which is important for both sensitivity and reliability estimation.

## F  VISUALIZATION RESULTS ON DRONEVEHICLE DATASET

**Visualization of Detection Results.** Fig. 7 presents qualitative comparisons of detection results on the DroneVehicle dataset. Each row corresponds to a distinct scene: the first column presents the ground truth annotations, followed by detection outputs from CALNet, CDC-YF, EI²Det, and our method. As observed, the baselines exhibit common errors, including misclassifications, missed detections, and false positives. In row (a), all baselines incorrectly classify a "freight car" as a "car" (red triangles); in row (b), the "truck" is entirely missed (yellow circles); in row (c), false positives appear around the "car" region (green triangles); and in row (d), another misclassification occurs, where the "freight car" is incorrectly labeled as a "truck." In contrast, our method enhances target recall, improves category discrimination, and mitigates the influence of cluttered backgrounds, enabling more robust and accurate detection in complex aerial scenarios with varying illumination.

**Visualization of Saliency Maps.** To further analyze how different methods allocate attention within the fused features, we visualize their saliency maps using LayerCAM (Jiang et al., 2021), as shown in Fig. 8. Each row corresponds to a distinct scene: the first column displays the ground truth annotations, while the remaining columns show the saliency maps generated by CALNet, CDC-YF, EI²Det, and our method, respectively. Across multiple scenes, the baseline methods consistently exhibit two types of attention failures: focusing on irrelevant regions (red dashed circles) and overlooking critical target regions (yellow dashed circles). In row (a), all methods fail to focus on the "tractor," and CDC-YF additionally neglects the "car," potentially leading to missed detections. In row (b), CALNet shows weak activation on the "freight car," while CDC-YF and EI²Det incorrectly attend to background regions, increasing the risk of false positives. In row (c), CALNet and CDC-YF are distracted by irrelevant regions, while CDC-YF and EI²Det fail to highlight the "car." In contrast, our method consistently allocates attention to relevant object regions while effectively suppressing background interference, resulting in more accurate localization and fewer detection errors. These results demonstrate that our method improves the reliability of extracted features, thereby enhancing multi-object detection performance under complex illumination conditions in aerial scenarios.

## G  THE USE OF LARGE LANGUAGE MODELS (LLMs)

In this study, we employed Large Language Models (LLMs) primarily for polishing the manuscript. The LLM was used to refine the text by correcting grammar and improving sentence structure. It is important to emphasize that the model did not contribute to the formulation of research ideas, the design of experiments, or the analysis of data.

The authors take full responsibility for the accuracy and integrity of the content and have reviewed all LLM-generated text to ensure it adheres to ethical standards, avoiding any potential issues such as plagiarism or misrepresentation.

