# OpenReview forum: "Reliability-Guided Gradient Correction for Visible-Infrared Object Detection"
_ICLR.cc/2026/Conference — ICLR 2026 Conference Withdrawn Submission_

### Official Review · Reviewer_d4ML · 2025-10-26

**Soundness:** 2
**Presentation:** 3
**Contribution:** 2
**Rating:** 4
**Confidence:** 3

**Summary:**

This paper tackles visible–infrared fused object detection. To better exploit object-relevant cues across modalities, the authors estimate a per-modality/region “reliability” signal from gradients and then resolve cross-modal conflicts by correcting gradients of less reliable modalities under the guidance of more reliable ones. Experiments on three datasets show improved detection performance.

**Strengths:**

- Visible-infrared object detection suffers from an inherent modality imbalance issue. Different modalities show complementary benefits, how to utilize these complementary benefits and resolve modalities' conflicts to enhance detection performance is a significant challenge.
- The proposed method is novel, estimating reliable regions/modalities during the fusion process with gradient information sounds interesting and reasonable, and utilizing the estimated “reliability” information to correct the “conflicts” between modalities is an efficient way to enhance performance without introducing additional computational cost during inference.

**Weaknesses:**

- Although it is interesting to leverage the “reliability” to enhance the multi-modal detection performance, the definition of the so called “reliability” is somewhat vague. In section 3.1.2, this paper proposes to define the “reliability” as the product of the normalized effectiveness score and the sensitivity score. This reads like an attention-style operation where the features important to the training objective are highlighted. Therefore, a more explicit discussion and comparison between the proposed method with attention mechanism would enhance the contribution of the gradient-based guidance.
- Several components appear heuristic (e.g., counterfactual reasoning for effectiveness, the precise conflict definition). The motivation and design rationality of these designs need further clarification.
- This paper lacks a comparison with some state-of-the-art methods. For example, ICAFusion (on multi-modal fusion strategy), CAGTDet (on handling misalignment), and QFDet (on sample assignment).
- In some settings, the improvements are modest given the complex operations introduced in the method.
- DronVehicle is a primary benchmark for the paper, while placing it in the appendix and showing weaker gains than on other datasets raises questions about generalization of the proposed method.
- Formatting issues and typos:
-- line 91, Repeated sentences
-- repeated references line 662.

[1] ICAFusion: Iterative Cross-Attention Guided Feature Fusion for Multispectral Object Detection, 2024.
[2] Improving RGB-infrared object detection with cascade alignment-guided transformer, 2024.
[3] Drone-based RGBT tiny person detection, 2023.

**Questions:**

See weaknesses

---

### Official Review · Reviewer_X3pK · 2025-10-31

**Soundness:** 3
**Presentation:** 3
**Contribution:** 3
**Rating:** 4
**Confidence:** 3

**Summary:**

This paper presents RaGrad, a model-agnostic approach designed to alleviate cross-modal semantic conflicts in visible–infrared object detection through reliability-guided gradient correction. The method first estimates the reliability of modality-specific parameters by jointly assessing their effectiveness and sensitivity, then refines unreliable gradients under the guidance of more reliable ones to promote consistent optimization. Experiments on multiple benchmarks demonstrate that RaGrad consistently improves detection accuracy and feature consistency across modalities with minimal computational overhead.

**Strengths:**

1.The paper presents a well-structured and mathematically grounded framework for visible–infrared object detection. The integration of reliability estimation and gradient correction is conceptually coherent and technically sound, forming a unified solution to mitigate cross-modal optimization inconsistencies. By quantifying the reliability of modality-specific parameters and guiding the optimization process accordingly, the method provides a principled and effective approach to improving multimodal feature learning.

2.The introduction of reliability as a guiding factor for gradient adjustment offers a novel and interpretable perspective on multimodal learning. This formulation transcends traditional heuristic fusion strategies by enabling adaptive, reliability-aware optimization that promotes stable and consistent feature alignment across modalities. The approach enhances both the robustness and explainability of multimodal fusion, representing a meaningful advancement in the field.

**Weaknesses:**

1.The computational efficiency of the reliability estimation process, particularly the counterfactual effectiveness computation and gradient variance analysis, is not sufficiently discussed. Although the authors claim minimal overhead, the practical cost of calculating reliability for large-scale backbones or high-resolution detection tasks could be significant. Providing quantitative runtime comparisons or complexity analysis would help substantiate the claim of scalability.

2.The reliability estimation relies heavily on gradient and parameter statistics, which may be sensitive to batch size, learning rate, and optimizer settings. The robustness of the reliability measure under varying training conditions is not systematically evaluated. Additional experiments or sensitivity analyses would help demonstrate the stability of the proposed mechanism in broader scenarios.

**Questions:**

1.How sensitive is the performance to the accuracy of the reliability estimation? For instance, what happens if reliability is computed less frequently or with coarser approximations?

2.Could reliability be incorporated as a prior during inference to enable adaptive modality weighting rather than only during training?

3.Could the assumption that more reliable gradients always provide the optimal optimization direction lead to over-correction or reduced gradient diversity in certain cases?

---

### Official Review · Reviewer_1kAE · 2025-10-31

**Soundness:** 1
**Presentation:** 2
**Contribution:** 1
**Rating:** 4
**Confidence:** 5

**Summary:**

This paper proposes a plug-and-play method to address semantic conflicts between infrared and visible modalities in object detection. The framework consists of a reliability estimation component and a reliability-guided conflict resolution module. The gradients are used to guide parameter updates according to the estimated reliability. The method is easy to apply, but its novelty is limited and the performance gains are not very significant.

**Strengths:**

1. The paper presents a plug-and-play solution that is simple to integrate into existing detection pipelines.
2. It introduces reliability estimation into infrared and visible object detection, which helps improve detection accuracy.
3. By using reliability to guide gradient-based parameter updates, the method offers better interpretability.

**Weaknesses:**

1. Uncertainty-aware learning and gradient calibration have been extensively studied; the proposed approach appears to be a straightforward combination of the two, which limits its originality.
2. The difference between the proposed uncertainty estimation scheme and existing attention-based mechanisms in this domain is not clearly articulated, and the advantages of the proposed method need further justification.
3. Given the rather marginal performance improvement, it is difficult to convincingly demonstrate the effectiveness of the proposed approach.
4. The model size and computational cost of the REPA and RGCR modules should be reported and evaluated to justify their necessity.

**Questions:**

1. The discussion of related work is not sufficiently thorough, and a more comprehensive analysis would better motivate the need for the proposed method.
2. The experimental comparison is limited, only two datasets are used, which weakly supports the claimed generalization ability.
3. In Fig. 1, the motivation and necessity of the proposed framework are not clearly conveyed; a clearer schematic is recommended.
4. The set of baselines is limited, and the visualizations do not include detection scores, which makes it hard to convincingly demonstrate the superiority of the method.

---

### Official Review · Reviewer_YShq · 2025-10-31

**Soundness:** 3
**Presentation:** 2
**Contribution:** 2
**Rating:** 4
**Confidence:** 4

**Summary:**

This paper tackles a fundamental challenge in visible–infrared object detection: the semantic inconsistency between RGB and IR modalities.

Existing methods mainly focus on feature alignment or heuristic weighting, assuming both modalities are equally reliable. However, these approaches fail when one modality produces noisy or misleading features under complex illumination or thermal conditions.

To overcome this limitation, the authors propose RaGrad, a novel reliability-guided gradient correction framework that explicitly models and utilizes the feature reliability of each modality during training. The method introduces two key components, which are 1) Reliability Estimation via Parameter Attribution (REPA) and 2) Reliability-Guided Conflict Resolution (RGCR).

RaGrad operates only during training, requires no architectural modification or additional inference cost, and is compatible with standard detectors such as YOLO and Faster R-CNN.

**Strengths:**

Strengths:
- Introduces reliability-guided gradient correction — a new perspective beyond feature alignment
- Works during training only, with no added inference cost or model modification
- Plug-and-play with existing detectors
- Lightweight, theoretically intuitive, and easily applicable to real multi-modal systems

**Weaknesses:**

Weaknesses:
- Reliability definition is not solid. The effect of parameters on prediction performance can be considered as modality reliability?
- No ablation studies on REPA and RGCR contributions
- Baseline models are outdated

**Questions:**

Please refer to the comments in weaknesses

---

### Note · Authors · 2025-12-10

I have read and agree with the venue's withdrawal policy on behalf of myself and my co-authors.